# Long-Term Comparative Study on the Local Tumour Control of Different Ablation Technologies in Primary and Secondary Liver Malignancies

**DOI:** 10.3390/jpm12030430

**Published:** 2022-03-09

**Authors:** Attila Kovács, Peter Bischoff, Hathal Haddad, Willi Zhou, Susanne Temming, Andreas Schäfer, Hannah Spallek, Lucas Kaupe, György Kovács, Michael Pinkawa

**Affiliations:** 1Clinic for Diagnostic and Interventional Radiology and Neuroradiology, Mediclin Robert Janker Klinik, 53129 Bonn, Germany; peter.bischoff@mediclin.de (P.B.); willi.zhou@mediclin.de (W.Z.); andreas.schaefer@mediclin.de (A.S.); hannah.spallek@rwth-aachen.de (H.S.); lucas.kaupe@semmelweis-hamburg.de (L.K.); 2Clinic for Radiotherapy and Radiooncology, Mediclin Robert Janker Klinik, 53129 Bonn, Germany; hathal.haddad@mediclin.de (H.H.); susanne.temming@mediclin.de (S.T.); michael.pinkawa@mediclin.de (M.P.); 3Gemelli-INTERACTS, Policlinico Universitario Agostino Gemelli IRCCS, Università Cattolica del Sacro Cuore, 00168 Rome, Italy; gyorgy.kovacs@unicatt.it

**Keywords:** interventional oncology, local ablative techniques, radiofrequency ablation, microwave ablation, cryoablation, electrochemotherapy, interstitial brachytherapy, local tumour control, personalised medicine

## Abstract

Purpose: To evaluate local tumour control (LTC) by local ablation techniques (LAT) in liver malignancies. Materials and methods: In patients treated with LAT between January 2013 and October 2020 target lesions were characterised by histology, dimensions in three spatial axes, volume, vascularisation and challenging (CL) location. LAT used were: Radiofrequency Ablation (RFA), Microwave Ablation (MWA), Cryoablation (CRYO), Electrochemotherapy (ECT), and Interstitial Brachytherapy (IBT). Results: 211 LAT were performed in 155 patients. Mean follow-up including MRI for all patients was 11 months. Lesions treated with ECT and IBT were significantly larger and significantly more often located in CL in comparison to RFA, MWA and CRYO. Best LTC (all data for 12 months are given below) resulted after RFA (93%), followed by ECT (81%), CRYO (70%), IBT (68%) and MWA (61%), and further, entity-related for HCC (93%), followed by CRC (83%) and BrC (72%), without statistically significant differences. LTC in hypovascular lesions was worse (64%), followed by intermediate (82% *p* = 0.01) and hypervascular lesions (92% *p* = 0.07). Neither diameter (<3 cm: 81%/3–6 cm: 74%/>6 cm: 70%), nor volume (<10 cm^3^: 80%/10–20 cm^3^: 86%/>20 cm^3^: 67%), nor CL (75% in CL vs. 80% in non CL) had a significant impact on LTC. In CL, best LTC resulted after ECT (76%) and IBT (76%). Conclusion: With suitable LAT, similarly good local tumour control can be achieved regardless of lesion size and location of the target.

## 1. Introduction

So-called “keyhole procedures” are constantly evolving and have become an indispensable part of modern medicine [1]. Today, image-guided local ablative therapies (LAT) expand the therapeutic spectrum in multimodal cancer therapy and open up new dimensions in terms of therapy tolerability and compliance [2]. Local tumour control (LTC) has established itself as an important pillar of modern oncology, complementing systemic chemotherapy (SCT) and surgery [1,2,3]. In the oligometastatic setting, patients can be provided long-term disease control and survival benefit if the critical foci of disease are removed. Thus, perhaps the most outstanding advantage of LAT is that it significantly increases overall survival (OS) in combination with SCT compared to SCT alone [4]. Surgery was and still is the gold standard for removing malignancies, now flanked by newer and less invasive methods such as stereotactic ablative radiotherapy (SABR) and a multitude of LAT [5]. The core goals of LAT are the achievement of local tumour and symptom control in the gentlest possible way, and with the shortest possible hospitalization [6].The choice of LAT depends on the parameters “entity, number, size and location” of the targets and, last but not least, on the expertise and armamentarium available on site [2]. Although the safety and effectiveness of the individual procedures have already been sufficiently evaluated, we also know today that not all procedures are equivalent in all situations [2,7]. However, modern personalised medicine requires the individualized use of LAT, adapted to the respective situation in the best possible way [1,8].

What is currently lacking is a comparative overview of all LAT available today, as well as a uniform algorithm for the use of the different techniques for image-guided ablation [3]. The authors have decades of experience with a broad spectrum of LAT. The aim of this paper is to systematically compare the newer percutaneous thermoablative, radioablative and chemoablative techniques—with respect to the above determinants of entity, number, size and location. In order to obtain comparable data, the study is limited to percutaneous local ablative procedures in the liver. For the same reason, endovascular locoregional procedures such as transarterial chemo- or radioembolization, as well as stereotactic radiation are not the focus of the study.

## 2. Material and Methods

### 2.1. Study Design

The retrospective analysis presented herein was designed to evaluate the LTC achieved with different LATs in primary and secondary liver malignancies. In all cases, LAT was performed for medical reasons, in accordance with the decision of a multidisciplinary tumour board. Institutional Review Board approval Ethics Commission University of Lübeck (AZ 22–027) was obtained, and all patients provided informed consent. The study was conducted in accordance with the European Union regulations, the Declaration of Helsinki, and the IHC Harmonized Tripartidute Guideline for Good Clinical Practice.

### 2.2. Patient Inclusion Criteria

Patients aged >18 years with histologically proven primary or secondary liver malignancies, assigned for LAT were considered eligible for the study. Patients were required to have liver-only or liver-dominant metastatic spread, with less than 25% of the liver parenchyma involved by tumour, and good performance status (ECOG-status 0–1; Eastern Cooperative Oncology Group).

### 2.3. Patient Exclusion Criteria

Patients with tumour invasion of the portal vein or the main biliary duct, portal vein thrombosis, serum bilirubin level higher than 2 mg/dL, transaminase values greater than three times upper limit of normal, renal insufficiency, contrast agent allergy and contraindication for sedation, were excluded.

## 3. Treatment Planning, Treatment Protocol and Follow-Up Assessment

Preinterventional planning was based on standardised multiparametric and contrast-enhanced magnetic resonance imaging (MRI) of the liver and computed tomography (CT). Based on this imaging, the target lesions were evaluated with regard to their morphological criteria. These included the maximum transverse diameter, the diameters in three spatial axes, the volume, the vascularisation (hyper-, intermediate- or hypovascularised), the segmental localisation, and the surroundings. The latter parameter captures the microenvironment of the target lesion, i.e., whether there are relevant structures nearby (>10 mm to the tumour margin), close (1–10 mm distance) or adjacent (<1 mm distance) that may make ablation challenging or affect the outcome of the ablation—e.g., main vessels or bile ducts. Challenging locations (CL) were defined as adjacent to central hepatic vessels, central bile ducts or hepatic hilum. All LAT, except ECT were performed under analgosedation and CT- fluroscopic image guidance. ECT alone was always performed under general anaesthesia. Meticulous planning was always carried out to determine how the patients had to be positioned in order to optimally cover the respective lesion with the planned ablation method. In this planning, great importance was attached to matching the size and the architecture of the target lesion with the geometry of the ablation probe/electrode, etc. The LAT-method was selected on the basis of the established parameters entity, size, volume, lesion geometry, localisation and environment of the target. In the case of previously treated lesions, the respective previous therapy was also taken into consideration.

Among the ablation methods used were

hyperthermic procedures:Radiofrequency Ablation (RFA/Boston Scientific/RFA3000^®^ Marlborough, MA, USA).Microwave Ablation (MWA/Terumo/TATO2^®^ Tokyo, Japan).hypothermic procedures: Cryoablation (CRYO/Galil/Boston Scientific/Visual Ice^TM^ Marlborough, MA, USA).chemoablative procedures: Electrochemotherapy (ECT/IGEA/Cliniporator vitae^®^ Modena, Italy).radioablative procedures: Interstitial Brachytherapy (IBT/VARIAN/GammaMed Plus^®^ Palo Alto, CA, USA).

Standard periinterventional medication consisted of cortisone and a single-shot antibiotic prophylaxis (500 mg Ciprofloxacin i.v.). In case of suspected infection antibiotic therapy was continued for two days via i.v. line and a further 5 days orally. Treatment related nausea and pain were treated with Ondansetron 4 mg/day and Ibuprofen 400 mg, each three times a day. Additionally, Piritramid (15 mg via i.v. perfusor) was available for the patients on demand.

The technical success was evaluated 48 h post intervention in MRI in terms of complete vs. incomplete ablation. Clinical and imaging follow-up with MRI was standardised at 1 month and then at 2-month intervals. Response to treatment was evaluated on the basis of the multiparametric MRI of the liver, including multiplanar ce T1w, transversal T2w fs and transversal DWI scans at aforementioned designated time points. Lesion-based treatment success (aforementioned parameter LTC) was assessed using the Modified Response Evaluation Criteria In Solid Tumours (mRECIST) criteria in terms of complete remission (CR), partial remission (PR), stable disease (SD), and progressive disease (PD). Local tumour control was defined as CR, PR or SD according to the RECIST criteria, version 1.1 [9]. Lesion-based progression free survival (PFS) was defined as the time interval between intervention and PD, or the last MRI imaging study performed, respectively. In the case of death, it was determined whether the death was causally related to the target lesion or not.

## 4. Statistical Analysis

Statistical analysis was performed using IBM SPSS Statistics 28.0 software (Armonk, NY, USA). An unpaired *t*-test was applied to compare age, follow-up, diameters and volumes between different treatments and a chi-square test was applied to compare categorial variables. The Kaplan-Meier methodology was used to assess local control and overall survival rates. The log-rank test was used to compare different interventions or other treatment-related factors. All *p*-values reported are two-sided; *p* < 0.05 is considered significant.

## 5. Results

Demographic patient characteristics are presented in Table 1. Overall, 211 minimal-invasive treatments were performed in 155 patients. Mean and median follow-up including MRI for all patients was 11 months and 7 months (up to 58 months). Patient age was similar between treatments. With exception of iBT, the majority of patients were female. Most frequent indications were CRC metastases in 40%, breast cancer metastases in 23% and hepatocellular cancer in 7%.

The majority of patients were on systemic treatment, with about a third with a prior liver surgery. ECT in particular was applied as a local lesion re-treatment in a large portion of patients. With only two exceptions before iBT, all treatment were based on a prior MRI for planning. Lesions treated with ECT and iBT were significantly larger in comparison to RFA, MWA and CRYO. Overall, 51% of lesions were located in challenging locations. Lesions treated with ECT and IBT were significantly more often located in challenging locations in comparison to RFA, MWA and CRYO.

As presented in Table 2, smaller lesions were treated predominantly using RFA and CRYO, while larger lesions were treated with iBT and ECT. Lesions in challenging locations were predominantly treated using iBT and ECT.

Considering all lesions (Figure 1), the best local control (LC) resulted after RFA (93% 12 months-LC), followed by ECT (81%), CRYO (70%), iBT (68%) and MWA (61%). LC after RFA was significantly higher in comparison to MWA (*p* < 0.01) and CRYO (*p* = 0.04). LC after iBT was significantly higher in comparison to MWA (*p* = 0.03).

Local control depending on primary tumour is presented in Figure 2. The best local control resulted for HCC (93% 12 months-LC), followed by CRC (83%) and BrC (72%), without statistically significant differences.

Additionally, local control was found to be significantly dependent on the vascular supply of the lesion (Figure 3). Follow-up of hypovascular lesions resulted in lower local control (64% 12 months-LC) in comparison to intermediate (82%; *p* = 0.01) or hypervascular lesions (92%; *p* = 0.07).

The maximum diameters and the volumes of the treated lesions showed different distributions. The volume divergence of the ECT target lesions was also more pronounced than that of the maximum diameter (Figure 4 and Figure 5). The greatest divergence in treated volume was seen for ECT (Figure 5).

Local control did not depend significantly on the maximum lesion diameter (<3 cm: 81% 12-month LTC; 3–6 cm: 74%; >6 cm: 70%) or lesion volume (<10 cm^3^: 80%; 10–20 cm^3^: 86%; >20 cm^3^: 67%), though a tendency for decreased local control resulted in larger lesions. Local control was not inferior for lesions in challenging locations (75% 12-month LTC vs. 80% in not challenging locations). In challenging locations, best local control resulted after RFA (83% 12 m-LC), ECT (76%) and iBT (76%); lower rates resulted after MWA and CRYO (67%, respectively; significantly lower comparing RFA vs. CRYO; *p* = 0.04).

Best overall survival (OS) after minimal-invasive treatment was reached after treatment for HCC (12 m OS 83%) in comparison to CRC (62%; *p* = 0.04), BrCa (64%; *p* = 0.09) and other primary tumours (50%; *p* = 0.04). The long-term 36 months OS was 88%, if only the liver was involved, 67%, if metastases were found in one other organ, and 42%, if metastases were found in more than one other organ at the time of first minimal-invasive treatment (difference statistically not significant).

Figure 6 shows an example of the response of a cooker treated with IBT. Images A1 and A2 are the pre-therapeutic images before local ablation, images B1 and B2 show the treated areas.

## 6. Discussion

In 2015, an international panel of renowned interventionalists met to develop recommendations for the use of LAT for primary and secondary liver malignancies [3]. For the recommendations, the advantages and disadvantages of each ablation technology were critically evaluated based on available data. As a conclusion of their evaluation, the authors called for clinical practice guidelines to include specific recommendations and protocols for individual techniques.

The single center study presented here intends to follow this recommendation and generate evidence from the perspective of clinical use, which can be utilized to make recommendations.

The study was conducted with the intention of comparing the effectiveness of different percutaneous and image-guided local ablative procedures. All investigated procedures have already proven their safety and effectiveness, and have already entered clinical routine worldwide. For better comparability, the study focused on ablations in only one organ, the liver. For the same reason, endovascular interventions e.g., transarterial chemo- or radioembolization, as well as stereotactic radiation were not the focus of the study. Among LAT, the classical thermoablative procedures such as RFA, MWA and cryoablation are represented. In addition, interstitial brachytherapy was included as a minimal invasive radioablative procedure and ECT as a chemoablative procedure. In order to be able to break down the local tumour control achieved with a particular procedure as structured as possible to the lesion characteristics, we have characterised the lesions in terms of their size and dimensions. This included the maximum diameter, the dimensions in three spatial directions and the volumetry. Further, the vascularization was assessed. Another aspect was to characterize the location of the respective lesion as precisely as possible. For this purpose, we used a scoring system that characterizes the microenvironment of the target lesion in parameters that are decisive for local ablative procedures. For example, this includes vessels in the immediate vicinity of the target lesion, which are known to lead to a heat sink effect.

The aim of our work was not to make a prospective randomized head-to-head comparison of all procedures. Just as the feasibility and safety of the individual procedures are known, so are their limitations. We have used this existing evidence to optimally apply the respective procedures according to current knowledge. Our motivation is also based on the fact that we have not just one, but numerous ablative procedures routinely in our institute and also use them in a targeted manner. Thus, for each intervention, a conscious decision was always made for a specific procedure from the entire portfolio of available technologies.

In our study, smaller lesions were treated predominantly using RFA and CRYO, while larger lesions were treated with IBT and ECT. Lesions treated with ECT and IBT were significantly larger in comparison to RFA, MWA and CRYO.

Similarly, lesions treated with ECT and iBT were significantly more commonly located in CL in comparison to RFA, MWA and CRYO. Lesions in CL were predominantly treated using iBT and ECT. Overall, and independent of the LAT used, 51% of lesions were located in challenging locations. This proportion is relatively high and can be explained by the fact that in MDTB, LAT is often recommended when the target lesions or the overall constellation is not suitable for surgical resection. In other words, it can be assumed that the surgically treatable lesions have already been selected [2].

A further interesting finding in our study is that ECT has been used most frequently as LAT for re-therapy of pretreated and locally recurrent lesions. The high proportion of recurrent lesions as targets in the ECT collective is supported by both the high proportion of CL and the larger volumes—in the latter case, recurrences were marginal recurrences, which are naturally associated with an increase in target volume. Chemoablation is an important addition to the LAT armamentarium. The good LTC that can be achieved with ECT further underpins its importance in the overall LAT portfolio.

Considering all lesions, the best LTC resulted after RFA, followed by ECT, CRYO, iBT, and MWA. Best LTC depending on the primary resulted for HCC, followed by CRC, and BrC without statistically significant differences.

Another interesting finding from our data is that LTC after LAT is also dependent on the vascularization of the target lesion—92% 12-month LTC in hypervascular vs. 64% in hypocascular lesions. This aspect, that vascularization is significantly relevant for the success of therapy, is so far only known from endovascular procedures such as transarterial chemoembolization (TACE) [10,11]. There, the situation is analogous–hypervascular lesions respond better to endovascular therapies. What is logical and understandable for endovascular therapies needs to be further explored and supported by more evidence for LAT.

In our study, LTC was not dependent on diameter or volume, nor on challenging location of the target. This can be explained by a suitable preselection-based on currently existing evidence—for the respective therapy that has been used [3].

The OS was dependent on the TU primary, respectively on the fact whether liver only disease was present and whether metastases were also found in one or more other organs.

Radiofrequency ablation (RFA) works with low-frequency and long radio waves (375–500 KHz), which lead to coagulative necrosis of the target cells in a closed circuit through frictional heat [2,3,6]. The volume of necrosis that RFA can produce is limited [3]. The main reason is carbonisation caused by excessive tissue desiccation, which increases resistance and reduces electrical current flow. Another reason is the heat sink effect in perivascular areas. As the oldest thermoablative procedure, RFA convinces with proven safety, reproducibility, and standardization [7]. Optimal results can be achieved with primary and secondary liver tumours up to a size of 3 cm. RFA is a classic single-probe procedure [1,3,7].

In microwave ablation (MWA), oscillating water molecules cause tissue heating. Compared to RFA, MWA systems operate in much higher frequency ranges (915 Hz–2.45 GHz) [2,5]. Because water, and not the tissue, is heated, MWA is also not limited by carbonisation. MWA is generally characterised by a faster propagation of heat, by higher temperatures (above the carbonisation threshold) and by an extensive independence from the heat-sink effect. Depending on the size of the target lesion, MWA is a single- or a multiprobe procedure. In larger lesions, several MWA antennas inserted in parallel can be operated synergistically. The physical advantages of MWA have not yet been translated one-to-one into clinical benefits [12]. Randomised comparisons have shown equivalent therapeutic effects and complication rates for MWA and RFA in the treatment of HCC, and colorectal liver metastases [12]. With regard to local tumour control, divergent and partly contradictory results have been published [13]. However, it is undisputed that MWA is faster than RFA [7]. Indications for MWA in the liver are HCCs with a size of up to 5 cm or for metastases up to 4 cm [2,5]. However limited LTC after MWA is more likely to occur in lesions larger than 3 cm in diameter, lesions located near large vessels and the diaphragm, as well as in chemo-resistant metastases, e.g., of colorectal cancer [14]. We currently have the widest variety of devices available in the field of MWA, so we have to assume limited predictability and reproducibility between brands, so our results are not necessarily applicable to all makes [13].

In cryoablation (CRYO), alternating freezing (up to −150 °C) and thawing (up to 40 °C) leads to the desired tumour necrosis [2,5]. Cell death is induced by intracellular and extracellular processes. Shock freezing of the cell organelles is achieved by the rapid and deep drop in temperature. Once the cell membrane is disrupted, extracellular fluid flows into the cell along the osmotic gradient, causing it to burst. Thrombotic vessel occlusion further leads to hypoxic necrosis [7]. Among thermal ablations, CRYO has the strongest immunological effect, promotes inflammatory cytokines, which also have a tumouricidal effect [15]. Advantages of CRYO are the low intra- and periprocedural pain, which allows it to be used on sensitive structures, the preservation of collagenous structures due to their resistance to cold, and the good visualisation of the frozen necrotic core in native CT [7]. CRYO can be used analogously to MWA as a single-probe procedure or, in the case of larger target lesions, synergistically as a multi-probe procedure. The disadvantages are the longer procedure time compared to MWA and the rare but severe systemic inflammatory reaction after ablation. This phenomenon, called cryoshock, was observed in the early days of open cryosurgery, mostly in large lesions. Evidence is lacking that these concerns are transferable to percutaneous CRYO [2,5]. In a prospective randomised clinical trial, percutaneous cryoablation produced the same 5-year overall survival rate as RFA in the treatment of HCC [2]. Due to the ease of intraprocedural monitoring of ice ball formation, CRYO is preferable in the vicinity of critical structures such as the diaphragm, heart, lungs, gallbladder and hepatic hilus, preferably for lesions smaller than 4 cm in diameter [2,5,7,16].

Electrochemotherapy (ECT) is a local chemoablation in which the anti-carcinogenic effect of the cytostatic drug is achieved by electroporation (EP) of the cell membrane [2]. Reversible EP is achieved by intra- and peritumoral electrodes positioned at regular intervals. The chemical substrate used is usually large-molecule, slow- to non-permeable drugs, such as bleomycin (BLM). Once inside the cell, the cytostatic causes multiple DNA breaks. To ensure complete and homogeneous coverage of the tumour volume, as well as adequate discharge of the electrodes, the applied electric field must be precisely planned and the electrodes must be positioned strictly parallel. The advantage of ECT is the controlled locoregional chemotherapy with a low cytostatic dose without pronounced systemic side effects. The targeted increase in cytotoxic effect (for BLM in vitro by a factor of 700) is achieved by EP [2]. The feasibility and safety of percutaneous ECT in hepatic malignancies has been demonstrated [17]. In patients with HCC, BLM-ECT achieved a complete response in 88.2% per lesion, and in patients with unresectable colorectal liver metastases in 85%. The analysis also showed that larger vessels (>5 mm) and bile ducts remained intact after ECT treatment [2]. Special features of ECT are the required general anaesthesia and muscle relaxation, because of the electrical impulses, as well as ECG synchronisation to ensure impulse delivery in the refractory phase of the heart. An increased risk of BLM-induced pulmonary fibrosis exists for elderly patients and in renal insufficiency. The maximum lifetime dose of BLM is 350 mg [2,5].

Interstitial brachytherapy (IBT) is an interventional local radioablation that uses high-dose remote afterloading (HDR) technology. In IBT, radiation sources (usually Ir192 with 370 GBq initial activity) are introduced via temporary, inactive applicators that have previously been placed in the defined target tissue in an image-guided manner. The special feature of IBT is that it is able to deliver a very high dose around the respective sources in the target volume. The surrounding normal tissue is largely spared, which is due to the steep dose drop outside the target volume [2,6,7]. The inactive IBT applicators are positioned CT- or sonography-guided. The aim is to achieve ideal target volume coverage. A major advantage of IBT is that, in contrast to the other multiprobe procedures, the applicators do not necessarily have to be positioned in parallel. Dose distribution is performed using a thin-slice CT scan with a suitable treatment planning software package (TPS). The clinical target volume (CTV), which includes gross tumour volume (GTV) and organs at risk (OAR), is delineated and a volume-optimised dose distribution is calculated. Typical single fraction doses are 15–20 Gy, with >50 Gy in the central (hypoxic and less radiosensitive) tumour region [2]. The duration of irradiation ranges from 20–60 min, depending on the activity of the iridium source and the size of the CTV. All modern remote HDR afterloading devices can be used for irradiation. IBT is a low-complication intervention. Special consideration must be given to the protection of radiosensitive organs such as the stomach or intestine [2]. IBT has some inherent advantages over thermal ablation techniques—for example, much larger target lesions with irregular architecture can be treated. Furthermore, the problem of heat dissipation does not occur with IBT. Therefore, peritumoral vessels do not interfere with the success of IBT [2]. In addition, IBT is superior to conventional radiotherapy (including stereotactic radiotherapy) in terms of the precision of local irradiation of the tumour while sparing the surrounding healthy tissue as much as possible [2]. IBT achieves excellent LTC of up to 96.1% in HCC and a high survival advantage over best supportive care (BSC) of 23 months mOS vs. 5 months [2]. In metastases, IBT also achieves promising LTC of 74.9–97.4%, depending on the primary entity (74.9–87.1% in CRC, 96.5–97.4% in breast cancer and 90% in liver metastases from pancreatic cancer) [2,6].

Minimally invasive ablations have increasingly complemented oncological surgery in recent years [7,8]. Interventional radiologists (IR) play an important role in interventional oncology (IO) and are an integral part of the multidisciplinary team in specialised centres. A basic prerequisite for the efficiency of multidisciplinary teams of specialists is a collaborative approach that, taken together, overcomes the limitations of single disciplines [2,7]. Patients for interventional treatments must be well selected. The weighing for and against certain therapies is reflected in the decisions of the multidisciplinary tumour board (MDTB) [6,7]. A significant improvement in overall survival has already been demonstrated in the long-term randomised CLOCC trial published in 2017 in patients with non-resectable colorectal cancer liver metastases (CRCLM) [4]. Analogous to the CLOCC trial, it is desirable to assess and expand the efficacy of LAT in the context of multimodal therapy concepts for other entities as well [18]. For patients in palliation, prolongation of life and quality of life are the decisive factors. In MDTB decisions, chemo holidays, shorter hospital stays and the associated shorter downtimes of patients are an increasingly important argument for IO [19,20,21,22]. Image-guided LAT have developed rapidly in the last decades, with significant technical improvements that have led to both an improved safety profile and better clinical outcomes [8,23,24,25]. From today’s perspective, the next requirements for LAT are to expand the indications, control the ablation field more precisely and achieve better long-term outcomes [5,26]. On par with technical innovations is evidence-based pre-selection of patients to achieve the best individualised outcomes [27,28]. From this point of view, it remains to be discussed whether the currently used diagnostic parameters are meaningful enough for this pre-selection [29,30,31,32].

In both interventional planning and the literature, the maximum diameter of the target lesion is currently the deciding factor for therapy [5]. This is correct as long as the target lesion and ablation zone are spherical. However, clinical experience shows that both malignancies, regardless of whether they are primary or secondary malignancies, and ablation volumes are primarily aspherical. In this respect, it is questionable whether the sole indication of the maximum diameter is sufficient to accurately plan an intervention, including the calculation of the ablative safety margin.

Furthermore, depending on the entity, the actual lesion borders cannot be captured equally well with all imaging modalities [8,18,32,33]. In colorectal cancer metastases, there is a discrepancy between the demarcation of lesion margins on contrast-enhanced CT and multiparametric MRI [11,24,28]. To achieve the best possible detection of tumour margins, we planned our study using both MR and CT imaging. Especially in intermediate or hypovascularised lesions, it is difficult to detect the margins with certainty. This may explain why LTC was worse in these lesions than in the hypovascularised lesions—it can be assumed that the tumour margin area has not been adequately ablated in hypovascular lesions. So far, it is only known for transarterial chemoembolisation (TACE) that hypovascularised liver metastases of colorectal carcinoma respond worse to therapy than hypervascularised metastases [10]. If the assumption is confirmed in future, that hypovascularised lesions are undersized, it will be essential to establish other imaging parameters for intervention planning in poorly demarcated lesions.

We used the ablation volumes specified by the manufacturers as a guide. In selecting the LAT, we followed the manufacturers’ specifications and used them to decide on the procedure to be used. In principle, when developing and introducing new ablation techniques, it would not only be advantageous but even necessary to establish real-time imaging controls for the individual procedures, as they already exist for the CRYO, for example. There is great hope that dedicated navigation systems, such as CASCINATION^®^ (Bern, Switzerland) and others, could provide not only better guidance but also better definition and thus coverage of the target lesions.

We agree with the call by Lencioni et al. that detailed information on technical parameters should be provided to allow a comprehensive understanding of the data and a critical assessment of the efficacy and safety profile of each ablation system. Ultimately, as opposed to a general indication of the technology, specific information on recommended techniques and protocols should be included in clinical practice guidelines, similar to pharmaceutical treatment regimens [3].

Complementary to local tumour and symptom control, the proven immunomodulatory effects of LAT will play an important, perhaps even more crucial, role than we now suspect, especially in the newly dawning era of checkpoint inhibitor therapy [15].

## 7. Conclusions

The size of the target lesion, as well as challenging localization, are not mandatory limitations for LAT. Our results show that even larger lesions can be treated effectively, provided that the limited ablation volume of individual electrodes and probes is respected, and multiprobe technologies are used that generate ablation volumes that better cover the target lesion. The same applies to targets in challenging localizations, e.g., in the immediate vicinity of large vessels. Here it is possible to use methods that are not limited by heat-sink effects, such as ECT or iBT.

As a result, it is advisable to have a broad spectrum of LAT available for a minimally invasive IO unit, which can be adapted to the individual situation. Another important aspect to overcome the limitations of unimodal approaches is interdisciplinary cooperation between all disciplines involved on the one hand to exploit synergistic therapeutic effects, on the other hand to find the timing of individual procedures in sequencing. According to Aristotle, the whole is more than the sum of the parts.

## Figures and Tables

**Figure 1 jpm-12-00430-f001:**
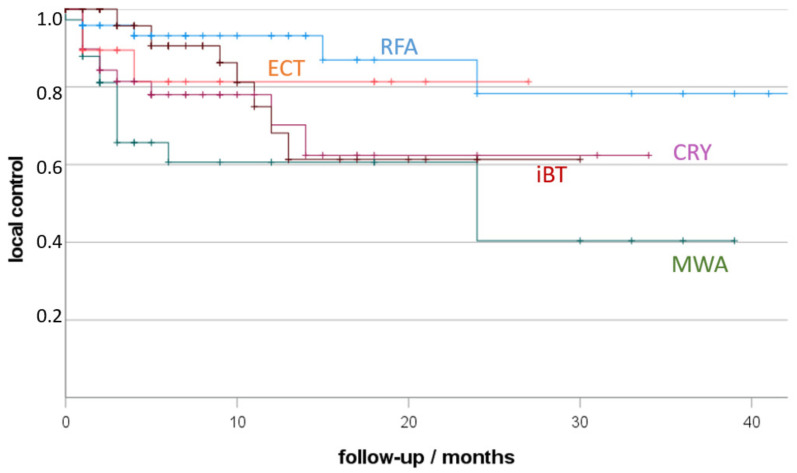
Local tumour control (LTC) depending on treatment (RFA, radiofrequency ablation; MWA, microwave ablation; CRY, cryoablation; ECT, electro-chemotherapy; iBT, interstitial brachytherapy; +, censored). Non-parametric Kaplan-Meier estimate of post-therapy LTC sorted by ablation procedure.

**Figure 2 jpm-12-00430-f002:**
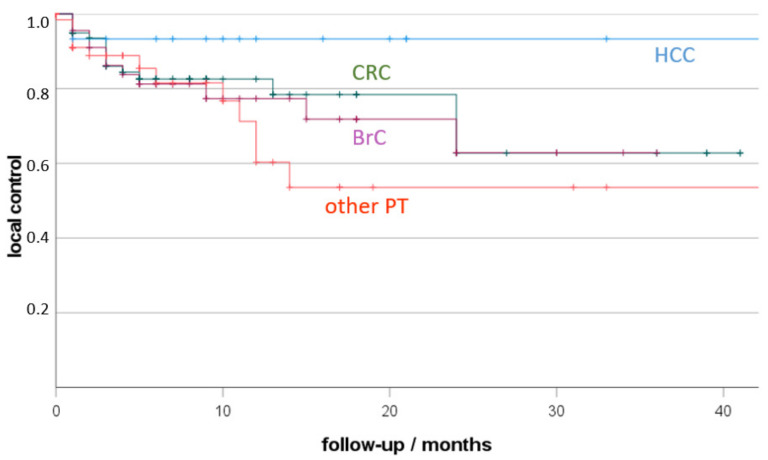
Local tumour control (LTC) depending on primary cancer (HCC, hepatocellular cancer, CRC, colorectal cancer; BrC: breast cancer; other PT: other primary tumour; +, censored). Non-parametric Kaplan-Meier estimate of post-therapeutic LTC sorted by malignancy.

**Figure 3 jpm-12-00430-f003:**
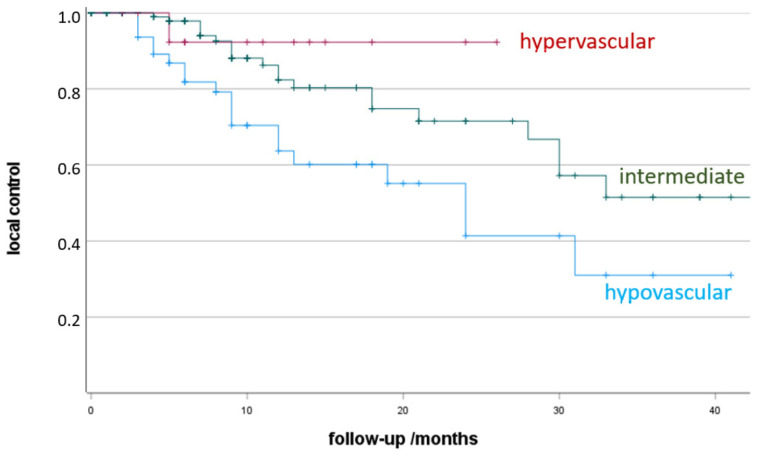
Local control depending on the vascular supply of the lesion (+, censored). Non-parametric Kaplan-Meier estimation of post-therapeutic LTC sorted by vascularisation of the target lesion.

**Figure 4 jpm-12-00430-f004:**
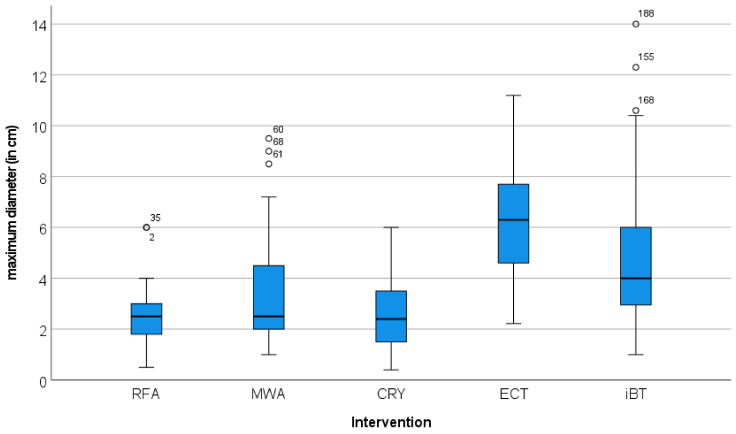
Distribution of the maximum diameter of the target lesions in relation to the local ablation procedure used. The graphical representation in the Box-Whisker-Plot shows comparable maximum diameters for the RFA-, MWA- and CRYO-treated lesions. The foci treated with ECT and iBT had significantly larger maximum diameters.

**Figure 5 jpm-12-00430-f005:**
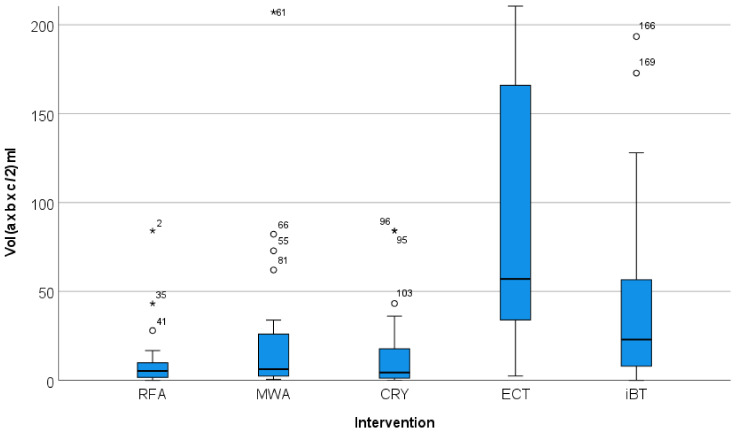
Distribution of the volume of the target lesions in relation to the local ablation procedure used (*, extreme value). The graphical representation in the Box-Whisker-Plot shows that the lesions treated with ECT were not only larger in median size than the lesions treated with RFA, MWA, CRYO and iBT, but also that they had a wider volume spread.

**Figure 6 jpm-12-00430-f006:**
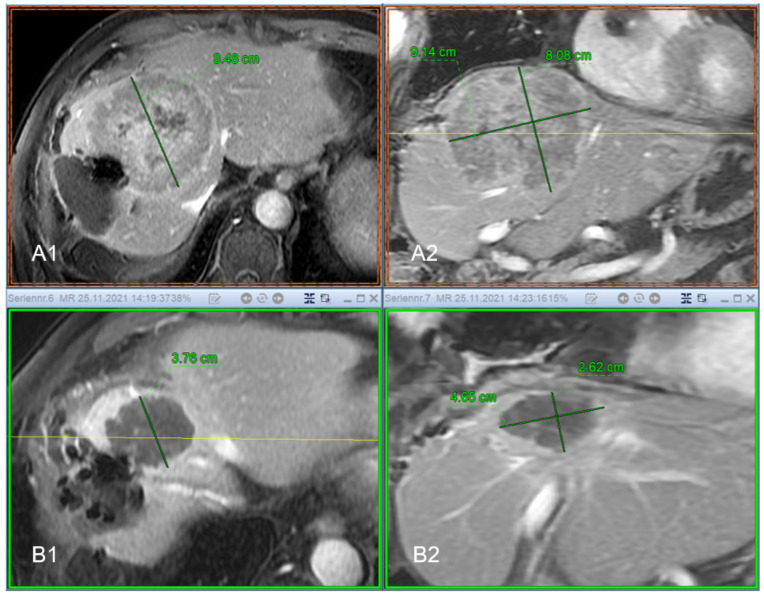
Seventy-three-year-old patient with previous surgery for Hepatocellular Carcinoma (HCC) and thus a significantly reduced liver reserve. The resection margin shows an extensive recurrence with the dimensions 8.48 cm × 9.14 cm × 8.08 cm (ap × lr × cc) and a volume of 313.13 cc ((**A1**) transversal contrast enhanced MRI, (**A2**) coronal contrast enhanced MRI, respectively). After successful radioablation by means of interstitial brachytherapy (IBT), there is a good response to therapy. In the imaging follow-up after 9 months, the lesion is completely necrotic, shows no signs of vitality and shrinks in time to 3.76 cm × 4.65 cm × 2.62 cm, which corresponds to a volume of 22.9 cc ((**B1**) transversal contrast enhanced MRI, (**B2**) coronal contrast enhanced MRI, respectively). Formally, the course of the findings is to be evaluated as a complete remission (CR).

**Table 1 jpm-12-00430-t001:** Demographic characteristics and aetiology of the treated malignancies, presence of multiorgan metastases, and characteristics of the target lesions summarised in a table and sorted by ablation procedure.

		RFA (*n* = 50)	MWA (*n* = 37)	CRYO (*n* = 42)	ECT (*n* = 21)	iBT (*n* = 61)
patient age/years (mean ± SD)		61 ± 13	63 ± 9	61 ± 13	62 ± 11	64 ± 12
median follow-up/months (MRI)		9	9	7	8	6
male		36%	41%	26%	48%	51%
indication	HCC	8%	8%	2%	14%	7%
	CRC	40%	35%	36%	38%	46%
	BrC	26%	11%	29%	24%	23%
	others	26%	46%	33%	24%	25%
other metastases	none	6%	8%	2%	14%	2%
	single organ	46%	28%	52%	33%	39%
	multiple organs	48%	46%	45%	52%	59%
systemic treatment		68%	70%	86%	86%	98%
hypovascular		22%	30%	48%	24%	34%
prior liver surgery		32%	30%	36%	24%	33%
prior local treatment		72%	81%	74%	62%	72%
lesion re-treatment		10%	33%	26%	43%	9%
max. diameter/cm (mean ± SD)		2.5 ± 1.1	3.4 ± 2.2	2.5 ± 1.4	6.6 ± 2.5	4.7 ± 2.9
volume/cm^3^ (mean ± SD)		8.2 ± 13	33 ± 71	12 ± 19	130 ± 137	67 ± 112

RFA, radiofrequency ablation; MWA, microwave ablation; CRYO, cryoablation; ECT, electro-chemotherapy; iBT, interstitial brachytherapy; HCC, hepatocellular cancer, CRC, colorectal cancer; BrC, breast cancer; SD, standard deviation; MRI, magnetic resonance imaging.

**Table 2 jpm-12-00430-t002:** Patient selection. Proportionate distribution of maximum lesion diameters, volumes and proportion of lesions in challenging locations tabulated and sorted by ablation procedure.

		Diameter			Volume		Challenging Location
	<3 cm	3–6 cm	>6 cm	<10 cm^3^	10–20 cm^3^	>20 cm^3^	
RFA (*n* = 50)	72%	28%	0%	76%	18%	6%	34%
MWA (*n* = 37)	57%	30%	13%	57%	16%	27%	35%
CRYO (*n* = 42)	64%	36%	0%	67%	14%	19%	48%
ECT (*n* = 21)	5%	38%	57%	5%	0%	95%	91%
iBT (*n* = 58)	25%	52%	23%	31%	14%	55%	63%

RFA, radiofrequency ablation; MWA, microwave ablation; CRYO, cryoablation; ECT, electro-chemotherapy; iBT, interstitial brachytherapy.

## Data Availability

Not applicable.

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
