# Peer review of "Long-Term Comparative Study on the Local Tumour Control of Different Ablation Technologies in Primary and Secondary Liver Malignancies"

_jpm, 2022, doi:10.3390/jpm12030430_

Round 1

Reviewer 1 Report

The paper deals with  the evaluation of local tumour control of liver malignancies with local ablation techniques. The topic of the manuscript refers to a significant health problem. The research meets all applicable standards for the ethics and research integrity. The article is presented in an intelligible fashion and adheres to appropriate reporting guidelines. Summary of results is given concisely in three tables and are clearly supported by accurate and good quality images. Conclusions are appropriate and the results as well as the discussion supports them.

Images are to be improved. Graphs have to be more clearly explained.

In discussion section please give more correlation to previously published comparative papers like "CT Appearance of Hepatocellular Carcinoma after Locoregional Treatments: A Comprehensive Review" by Marin et al.

Enlish need some revision, and editing is needed.

Well done!

Author Response

Dear Ladies and Gentlemen, dear Colleagues,

thank you for the detailed statement and for the informative and constructive suggestions - all of which we have worked through and implemented as requested. The manuscript has been extensively revised according to the reviewers' wishes - unfortunately it took more time than estimated. However, we hope that we have now fulfilled all the requirements.

The changes (in italics) in detail are as follows

Many thanks for the support

Images are to be improved. - images have been completely edited and are now also available separately as a high-resolution file.

Graphs have to be more clearly explained - the legends have been edited

In discussion section please give more correlation to previously published comparative papers like "CT Appearance of Hepatocellular Carcinoma after Locoregional Treatments: A Comprehensive Review" by Marin et al. – Thank you for the hint. The reference has been added.

Enlish need some revision, and editing is needed – Thank you for the tip. The complete article has been reworded and edited - it is now shorter and more precise overall.

Reviewer 2 Report

General comments:

Interesting article addressing multimodality ablation of liver tumor.

The authors report outcomes with radiofrequency ablation that is significantly greater than microwave.  This is discordant with established literature and should be addressed/explained.

Multiple run-on sentences that make the manuscript difficult to read and need to be rephrased.

Specific comments:

Numerous abbreviations are used throughout the manuscript.  I would recommend an abbreviation list near the beginning of the manuscript.

I don’t feel that 18 citations are enough considering the content of this manuscript with multiple modalities of ablation treatment addressed.

When do you utilize embolic therapy such as TACE, DEB-TACE, and radioembolization with yttrium 90 in the treatment algorithm? The reader would be led to believe that ablative therapy is utilized at the author’s institution no matter the size and location of tumor in the liver, which would be an unusual approach.

Page 4: “…or adjacent (>1 mm)…” should be < 1 mm.

Page 7 Figure 4. I don’t understand “maximeter Durchmesser”

If the mean f/u for patients was 11 months with imaging, how were you able to report local control out to 40 months in figures 2 and 3 as well as report OS of 36 months on page 8?

Page 8 Figure 6 legend: I don’t understand “vitality and shcrumps”

Page 8 first paragraph. I don’t understand 12m-LC regarding local control reporting.  This is also stated at the bottom of page 7.

Page 9 “Conversely, no intervention was carried out of the embarrassment of…”  This sentence is unclear.  Please rephrase and avoid use of the word embarrassment.

Page 13 “IO’s can convince with lower costs.”  This is unclear.  Please rephrase.

Page 10 RFA: “with a long wavelength to determine heat…”  This is unclear.  Please rephrase.

Page 10 RFA: “till 3 cm in size.” Would rephrase to under or less than 3 cm in size.

Page 11 “…the tolerance of CRYO is very good due to the low pain.”  Would rephrase “…the tolerance of CRYO is very good due to less pain.”

Page 11 “…further the voltage for each couple of electrodes.”  This sentence is unclear.  Please rephrase.

Author Response

Dear Ladies and Gentlemen, dear Colleagues,

thank you for the detailed statement and for the informative and constructive suggestions - all of which we have worked through and implemented as requested. The manuscript has been extensively revised according to the reviewers' wishes - unfortunately it took more time than estimated. However, we hope that we have now fulfilled all the requirements.

The changes (in italics) in detail are as follows

Many thanks for the support

The authors report outcomes with radiofrequency ablation that is significantly greater than microwave.  This is discordant with established literature and should be addressed/explained. - Thank you for pointing out the performance of the MWA. Our literature search has revealed a divergent picture of the performance of both procedures, MWA and RFA. Randomised comparisons show comparable therapeutic effects and complication rates of MWA and RFA in the treatment of HCC and colorectal liver metastases. With regard to local tumour control, divergent and partly contradictory results have been published - which are also discussed in meta-analyses as issues that still need to be clarified. In fact, however, we also expected better performance from MWA - but this may also be due to our device. Currently, MWA technology has the largest variety of available devices on the market, so we have to assume limited predictability and reproducibility between brands, so our results are not necessarily transferable to all makes. Even in the early stages of MWA technology, it was suspected that different brands performed differently - this was proven in a study published in Radiology in 2013.

Multiple run-on sentences that make the manuscript difficult to read and need to be rephrased. - Thank you for the tip. The complete article has been reworded and edited - it is now shorter and more precise overall.

Numerous abbreviations are used throughout the manuscript.  I would recommend an abbreviation list near the beginning of the manuscript. - included

I don’t feel that 18 citations are enough considering the content of this manuscript with multiple modalities of ablation treatment addressed. - Thank you for the tip. Missing references have been added.

When do you utilize embolic therapy such as TACE, DEB-TACE, and radioembolization with yttrium 90 in the treatment algorithm? The reader would be led to believe that ablative therapy is utilized at the author’s institution no matter the size and location of tumor in the liver, which would be an unusual approach. – Thank you for pointing this out. This aspect has now been clarified in the introduction as well as in the discussion. In order to obtain comparable data, the study is limited to percutaneous local ablative procedures in the liver. For the same reason, endovascular locoregional procedures such as transarterial chemo- or radioembolisation as well as stereotactic radiation are not the focus of the study. The results of endovascular therapies have already been published elsewhere, at least in part. Seidl S, Bischoff P, Schaefer A, Esser M, Janzen V, Kovács A. TACE in colorectal liver metastases – different outcomes in right-sided and left-sided primary tumour location. Integr Cancer Sci Ther 2020; 7(1)

Page 4: “…or adjacent (>1 mm)…” should be < 1 mm - thank you, corrected

Page 7 Figure 4. I don’t understand “maximeter Durchmesser” - thank you, reworded

If the mean f/u for patients was 11 months with imaging, how were you able to report local control out to 40 months in figures 2 and 3 as well as report OS of 36 months on page 8? – thank you; follow-up for our patients ranged up to 58 months. Therefore, local control in figures 2 and 3 show follow-up periods >40months and OS of 36 months could be reported in the manuscript.

Page 8 Figure 6 legend: I don’t understand “vitality and shcrumps” – thank you, reworded

Page 8 first paragraph. I don’t understand 12m-LC regarding local control reporting.  This is also stated at the bottom of page 7. – thank you, reworded

Page 9 “Conversely, no intervention was carried out of the embarrassment of…”  This sentence is unclear.  Please rephrase and avoid use of the word embarrassment. - Thank you, indeed was oddly worded. It has now been reworded accordingly. Thus, for each intervention, a conscious decision was always made for a specific procedure from the entire portfolio of available technologies.

Page 13 “IO’s can convince with lower costs.”  This is unclear.  Please rephrase. - Thank you - I have decided to delete the sentence completely because it makes no relevant contribution to the current topic.

Page 10 RFA: “with a long wavelength to determine heat…”  This is unclear.  Please rephrase. – Thank you. I have rephrased the sentence. Radiofrequency ablation (RFA) works with low-frequency and long radio waves (375 - 500 KHz), which lead to coagulative necrosis of the target cells in a closed circuit through frictional heat.

Page 10 RFA: “till 3 cm in size.” Would rephrase to under or less than 3 cm in size. - Thank you. I have rephrased the sentence. Optimal results can be achieved with primary and secondary liver tumours up to a size of 3 cm.

Page 11 “…the tolerance of CRYO is very good due to the low pain.”  Would rephrase “…the tolerance of CRYO is very good due to less pain.” - Thank you - I have shortened the entire paragraph and rephrased it as follows: Advantages of CRYO are the low intra- and periprocedural pain, which allows it to be used on sensitive structures, the preservation of collagenous structures due to their resistance to cold, and the good visualisation of the frozen necrotic core in native CT.

Page 11 “…further the voltage for each couple of electrodes.”  This sentence is unclear.  Please rephrase. - Thank you - I have also completely reworded this paragraph: To ensure complete and homogeneous coverage of the tumour volume, as well as adequate discharge of the electrodes, the applied electric field must be precisely planned and the electrodes must be positioned strictly parallel.

Round 2

Reviewer 1 Report

All tasks were accomplished